# Ki-67 as a Prognostic Biomarker in Invasive Breast Cancer

**DOI:** 10.3390/cancers13174455

**Published:** 2021-09-03

**Authors:** Matthew G. Davey, Sean O. Hynes, Michael J. Kerin, Nicola Miller, Aoife J. Lowery

**Affiliations:** 1Discipline of Surgery, The Lambe Institute for Translational Research, National University of Ireland, H91 YR71 Galway, Ireland; michael.kerin@nuigalway.ie (M.J.K.); nicola.miller@nuigalway.ie (N.M.); aoife.lowery@nuigalway.ie (A.J.L.); 2Department of Surgery, Galway University Hospitals, H91 YR71 Galway, Ireland; 3Department of Histopathology, National University of Ireland, H91 YR71 Galway, Ireland; Sean.hynes@nuigalway.ie

**Keywords:** breast cancer, biomarker, Ki-67, MIB-1, personalised medicine

## Abstract

**Simple Summary:**

In breast cancer development, the expression of Ki-67 is strongly associated with cancer proliferation and is a known indicator of prognosis and outcome. Ki-67 expression levels are also useful to inform treatment decision making in some cases. As a result, routine measurement of Ki-67 is now widely performed during pathological tumour evaluation. However, the Ki-67 appraisal is not without its limitations and shortcomings—the aim of this study was to provide an overview of Ki-67 use in the clinical setting, the current challenges associated with its measurement, and the novel strategies that will hopefully enhance Ki-67 proliferation indices for prospective breast cancer patients.

**Abstract:**

The advent of molecular medicine has transformed breast cancer management. Breast cancer is now recognised as a heterogenous disease with varied morphology, molecular features, tumour behaviour, and response to therapeutic strategies. These parameters are underpinned by a combination of genomic and immunohistochemical tumour factors, with estrogen receptor (ER) status, progesterone receptor (PgR) status, human epidermal growth factor receptor-2 (HER2) status, Ki-67 proliferation indices, and multigene panels all playing a contributive role in the substratification, prognostication and personalization of treatment modalities for each case. The expression of Ki-67 is strongly linked to tumour cell proliferation and growth and is routinely evaluated as a proliferation marker. This review will discuss the clinical utility, current pitfalls, and promising strategies to augment Ki-67 proliferation indices in future breast oncology.

## 1. Introduction

### Biomarkers

The biomolecular era, initiated by Crick, Franklin, and Watson following their precise description of the structure of deoxyribose nucleic acid in 1953, led to a substantial expansion of our understanding of the molecular basis of disease and the subsequent utility of biomarkers in clinical practice. A biomarker, a portmanteau of ‘biological marker’, is a characteristic that is objectively measured as an indicator of normal biological processes, pathological processes, pharmacological responses to a therapeutic intervention [1], or to predict incidence or outcome of disease [2]. Biomarkers are used to provide information concerning human biology, and the development of novel oncological biomarkers remains at the forefront of translation research priorities. There are several categories of biomarkers; diagnostic biomarkers are used to distinguish diseased from healthy individuals, while predictive, prognostic and therapeutic biomarkers may influence therapeutic decision-making and management strategies with the aim of personalising disease treatment [3,4]. Prognostic biomarkers focus upon identifying the likelihood of a clinical event in the setting of disease [5]. Unfortunately, sometimes prognostic biomarkers are a blunt measure of stratifying outcomes, and their reliability is limited through interindividual variability (i.e., differing values for a spectrum of patients), intraindividual variability (i.e., differing scoring by histopathologists providing Ki-67 measurement), and sensitivity and specificity implications [3]. Consequentially, biomarkers are not always absolute in predicting outcomes. 

Breast cancer is responsible for 1.7 million new cancer diagnoses worldwide each year [6]. Traditionally, breast cancer was considered a homogenous entity [7], with radical resection through mastectomy the cornerstone of effective cancer control [8]. The molecular era has transformed breast cancer management [9]: We now consider invasive breast carcinoma a heterogenous disease with varied morphology, tumour behaviour, response to therapeutics and molecular features [10]. Furthermore, the discovery and development of diagnostic, prognostic and therapeutic biomarkers have transformed the international perception such that at least four heterogeneous molecular subtypes are recognised in clinical practice [11,12]. Distinguishing these subtypes relies on the genetic expression of estrogen receptor (ER) status, progesterone receptor (PgR) status, human epidermal growth factor receptor-2 (HER2) status, and Ki-67 proliferation indices due to their critical role in the substratification, prognostication, and personalization of treatment modalities for each subtype [10,12,13,14,15,16,17,18,19]. Mandatory ER, PgR, and HER2 immunohistochemical appraisals are recommended to approximate the genetic expression of these in all cases of invasive breast cancer according to the American Society of Clinical Oncology/College of American Pathologists (ASCO/CAP) guidelines [20,21], as these are established predictive and prognostic biomarkers in breast oncology, proving crucial in therapeutic decision making [18,22,23,24]. Additionally, Ki-67 proliferation indices remain critical in the 2011 St. Gallen Consensus for differentiating Luminal A and Luminal B molecular subtypes [12]. Ki-67 expression is strongly associated with aggressive tumour biology and tumour proliferation, and recognition has grown for Ki-67 as an excellent prognostic biomarker [25,26].

Currently, certain authors report the inherent value of Ki-67 in breast oncology [27], while controversy exists as to the reliability of Ki-67 in independently predicting responses to therapy and survival. This review will focus on the current clinical utility of Ki-67 indices, highlight current pitfalls of the biomarker, and outline strategies that may enhance Ki-67 application in future practice.

## 2. Ki-67 Proliferation Indices

Antigen Ki-67, also known as Ki-67 or Marker of Proliferation Ki-67 (MKI67), is a protein in humans encoded by the MKI67 gene [28]. Ki-67 encodes two protein isoforms with molecular weights of 345 and 395 kilodaltons and was initially identified in Hodgkin lymphoma cell nuclei 1983 by Gerdes and Scholzer [29]. The name of this biomarker is derived from its city of origin, Kiel, and its location within the 96-well plate [30]. The quantity of Ki-67 present at any time during the cell cycle is regulated by a precise balance between synthesis and degradation, as indicated through its short half-life of 60–90 minutes [31,32]. Ki-67 remains active during the G1, S, G2, and M phases of the cell cycle [33], making it an excellent marker of cell proliferation [34,35] and an accepted hallmark of oncogenesis [36]. During interphase, the Ki-67 antigen can be exclusively detected within cell nuclei, whereas in mitosis, most of the protein is relocated to the surface of cellular chromosomes [37]. Ki-67 remains absent during the quiescent G0 phase, and levels reduce significantly during anaphase and telophase [38]. Immunohistochemical evaluation of Ki-67 is now incorporated into the paradigm for several cancer types due to its reliable correlation with the proliferative activity of cancer cells [39]. Reliable prognostication using Ki-67 as a solitary biomarker has been validated in a number of cancers, including breast, prostate, cervical, lung, soft tissue, neuroendocrine cancers, and gastrointestinal stromal tumours [40,41,42,43,44,45]. In contemporary clinical practice, Ki-67 is often considered a reliable indicator of responses to systemic therapeutic strategies and acts as a prognostic biomarker in certain malignancies [46,47]. In spite of this, difficulties surrounding the evaluation, utilisation, and credibility of Ki-67 have hampered the uniform uptake of Ki-67 in routine practice.

### 2.1. Ki-67: Inconsistencies in Detection

Extensive efforts have been made over the past three decades to evaluate the predictive value of the Ki-67 proliferation index [48,49,50]. In spite of these endeavours, this biomarker has not been completely integrated as a standard component of clinical decision making or pathological reporting [51], largely due to the difficulty standardising methods of Ki-67 appraisal [25,52]. As outlined by the International Ki-67 Working Group [52], inconsistencies in scoring are possible at the preanalytical, analytical, interpretation, and data analytic phases of Ki-67 evaluation. During the preanalytical phase, a number of parameters could all potentially contribute to differences in the assessment of Ki-67. These include specimen type, fixative type, cold ischaemic time (i.e., time taken from the removal of the specimen at surgery to the placement to the fixation of the tissue), as well as the length of fixation [53]. Although it has been shown that fixation for up to 154 days may not negatively impact Ki-67 staining, in practice, the standard methods used for fixation, i.e., buffered formalin as a fixative for 8–72 hours, are adequate for ensuring accurate Ki-67 results [21]. The type of specimen, such as cytology or histology, could potentially lead to differences in Ki-67 scoring as they utilised different fixatives. Another important practical consideration is the surgical procedure. A mastectomy can produce significantly more tissue than a wide local excision, which, if not correctly handled, may prevent fixation of central tumour tissue as formalin has a penetrance of mm per hour. A lack of fixation increases the cold ischaemic time and can cause cells to drop out of the cell cycle, decreasing Ki-67 scores [52,52,54]. However, standard histopathology tissue handling practices, in general, prevent this from occurring. Moreover, following processing and embedding, the tissue remains stable in a paraffin block for a longer time than a cut section, and so freshly cut sections should always be used for a standard assessment approach.

In the setting of immunohistochemistry analysis, antigen retrieval, antibody selection, colorimetric detection, as well as adequate counterstaining of the negative nuclei all require standardisation to ensure the clinical reliability of Ki-67, which will be the case in a clinically accredited laboratory. Misinterpretations of scoring may lead to inconsistencies in Ki-67 reporting; this may occur through interpersonal variability. Controversary surrounding data analysis within Ki-67 is apparent due to the lack of recommended consensus guidelines, with uncertainty surrounding the selection of relevant cut-off points for this biomarker. Furthermore, there are several means of staining and evaluating Ki-67, which can potentially lead to inconsistencies in scoring, while variability in interlaboratory methodology can also limit the reproducibility of this biomarker. For example, cytoplasmic staining and occasional membrane staining of breast cancer cells for the Ki-67 antigen can occur with the MIB1 antibody [20], although only nuclear staining (plus mitotic figures) should be included in Ki-67 scoring. The Ki-67 score is defined as the percentage of positively stained cells among the total number of cancer cells assessed [55]. In using MIB1 staining, probably the single most confounding factor in accurate assessment is the heterogeneity of expression. The gradient of increasing staining between tumour hot spots and tumour peripheries (the leading edge is expected to the most biologically active site of the tumour) can cause difficulties in judging where is most representative of the tumour overall. Currently, MIB1 is the most commonly used clone for Ki-67 appraisal [56] and has built up a long and validated track record, making it considered by many as the ‘gold standard’ [52,57]. However other clones can be used and these include: SP6, 30-9, K2, and MM1. [58,59,60,61]. Interestingly, the rabbit anti-human Ki-67 monoclonal antibody SP6 recognises the identical repeated Ki-67 epitope as MIB1 and looks promising to enhance sensitivity for quantitative image analysis [62,63], as validated in several recent studies [64,65]. The most recent edition of the American Joint Committee on Cancer (AJCC) describes recommendations relating to the routine measurement of Ki-67 expression as ‘AJCC Level of Evidence: III’, encapsulating the variability of this biomarker in histopathological cancer staging.

### 2.2. Ki-67 Guidelines and Therapeutic Decision Making

#### 2.2.1. Ki-67 Clinical Interpretation

Existing guidelines are inconsistent with regard to interpreting clinically relevant cut-off points in Ki-67 expression: In 2011, the 12th St. Gallen Expert Consensus panel established a measurement of less than 14% in ER positive (ER+) disease to represent the Luminal A molecular subtype, while scores in excess of this fitted with the Luminal B (HER2 negative) molecular class [12]. Updates from the 2013 St. Gallen consensus statement redefined greater than 20% as the new threshold for substratifying Luminal subtypes [66] based on the work of Prat et al., which highlighted the relevance of this cut-off to predict survival outcomes within the ER+ cohort [10,67]. This shift in the accepted threshold was modelled from data suggesting tumours with a greater Ki-67 expression were more likely to benefit from cytotoxic chemotherapy. Additionally, Enrico et al. described an optimal cut-off of 23.4% for differentiating Luminal breast cancer molecular subtypes [68], following validation in 506 stage I–III breast cancer patients in 2018. Although this is somewhat of an unrealistic conventional cut-off, the authors also highlighted a 20% cut off to be clinically relevant for recurrence and survival (hazard ratio (HR): 7.14, 95% confidence interval (CI): 3.87–13.16). Furthermore, Petrelli et al. outlined the prognostic significance of Ki-67 expression levels greater than or equal to 25% for predicting mortality in their review of over 64,000 breast cancer patients (HR: 2.05, 95% CI: 1.66–2.53) [69]. More recently, Tian et al. describe Ki-67 utilisation in isolation as valid for those with scores less than 15% and greater than 30%, with patients with borderline scores falling between these values best supplemented with the 70-gene (MammoPrint) or 80-gene signatures (BluePrint) [70]. Of note, the rate of miscalculation of Ki-67 was just 11% in cases carrying expression less than 15% or greater than 30%; hence, their conclusions implying genomic testing should augment therapeutic decision making in this group. Zhu et al. also suggested a cut-off of 30% to be clinically relevant in ‘de-escalating’ aggressive systemic therapy prescription in their series of 1800 triple negative breast cancer (TNBC) cases [27]. Baseline levels of Ki-67 expression in TNBC are expected to be higher than in Luminal tumours [71], and definitions of cut-offs within triple negative disease are diverse and inconsistent, withreported values of as low as 10% and as high as 35% within TNBC disease [72,73], and a recent meta-analysis of 35 independent studies of almost 8000 patients with resected TNBC suggests a cut-off of 40% is associated with a greatest risk of disease recurrence and mortality [74] (Table 1).

#### 2.2.2. Ki-67 Guidelines

The current guidelines surrounding Ki-67 and its role in therapeutic decision making are controversial: The most recent update from the International Ki-67 Working Group accepted Ki-67 as a prognostic marker in breast carcinoma, however, concluded that clinical utility is evident only for prognostic estimation in Luminal disease to guide therapeutic decision-making regarding adjuvant chemotherapy prescription. Additionally, consensus suggests that Ki-67 ≤ 5% or ≥30% can be useful in estimating prognosis in early-stage, luminal disease [52]. This is congruent with previous guidelines: In 2016, ASCO released clinical practice guidelines, which distinctly outlined that the ‘Protein encoded by the MKI67 gene labelling index by immunohistochemistry should not be used to guide choice on adjuvant chemotherapy’, and hesitancy in relying upon ‘Ki-67 protein levels in tumour cells to make recommendations about the type of hormonal therapy prescribed after surgery’, as well as ‘cancer cells with high levels of Ki-67 don’t respond well to aromatase inhibitors’ [22]. These recommendations, derived from studies of intermediate levels of evidence, added further to the ambiguity of Ki-67 evaluation in clinical practice. Moreover, the moderate strength of recommendation in relation to implementing these guidelines added even further obfuscation [22]. Furthermore, there has been recent evidence highlighting the Ki-67 score observed on core biopsy is systematically different from those observed on the excised cancer specimen, limiting the consistency of the biomarkers’ utility in certain settings [78].

In order to address some of these challenges, the International Ki-67 Working Group has developed a systemic multiphase program assessing whether Ki-67 scoring may be analytically standardised and validated across laboratories worldwide [52,79,80]. Phase I studies illustrated substantial interobserver variability among some of the world-leading experts in breast pathology on TMA of whole tissue specimens [79], while phase II reduced variability by applying a standardised, practical visual scoring method [80]. Furthermore, the phase III study demonstrated that it is possible for pathologists to achieve high interobserver agreement in scoring Ki-67 on cut biopsies using only a conventional light microscope and manual field selection [81]. This was achieved using the scoring system validated in the phase II study [80].

#### 2.2.3. Ki-67 and Endocrine Therapies

In 2015, the International Ki-67 Working Group provided an update concerning the validity of utilising Ki-67 as a clinical marker of response to neoadjuvant therapies [82]: In neoadjuvant endocrine therapies (NAET), Ki-67 is a predictive biomarker of response and long-term clinical outcomes, hence its inclusion in several prospective trials evaluating response to NAET in breast carcinoma, including the Immediate Preoperative Anastrozole, Tamoxifen, or Combined with Tamoxifen (IMPACT), and Arimidex, Tamoxifen Alone, or Combined (ATAC) trials [83,84,85,86]. A recent meta-analysis illustrated the value of the 21-gene assay (which includes Ki-67) as a valuable tool in predicting response to NAET [87]. Moreover, Ki-67 has been assessed as a marker to substratify patients with partial response to neoadjuvant chemotherapy (NAC) who may require extended systemic therapy due to a higher predicted risk of relapse and those who can proceed to primary surgery [88]. Residual cancer burden has been identified as correlative to long-term clinical outcomes following NAC in breast cancer patients, and increased Ki-67 in the interim between finishing NAC and undergoing resection indicates poorer outcomes [89,90,91]. In spite of this explicit prognostic information, Ki-67 measurement remains inconsistent and irreproducible between patients, limiting updates to current guidelines surrounding the routine inclusion of Ki-67 staining in standard breast cancer immuno-histochemical workup.

#### 2.2.4. Ki-67 and Triple Negative Breast Cancer

The introduction of immune checkpoint inhibitors (ICIs) into breast oncology has been limited when compared to other cancers such as non-small cell lung (NSCLC), malignant melanoma, bladder and rectal cancers [92,93,94,95]. Currently, the IMPASSION 130 and Keynote 522 trials indicate promise with respect to the role of ICIs in treating TNBC in the early-stage (HR: 0.63, 95% CI: 0.43–0.99) and metastatic settings (HR: 0.80, 95% CI: 0.69–0.92) for combined ICI and conventional chemotherapy compared to placebo and chemotherapy [96,97]. Programmed cell-death ligand 1 (PD-L1, B7-H1 or CD274), the complimentary ligand of Programmed cell-death 1 (PD-1), is expressed on the surface of cancer cells and recruited immune cells and suppresses the local immunological response to cancer cells by inducing apoptosis of tumour infiltrating lymphocytes (TILs), leading to propagation of tumour proliferation. Consequentially, high PD-L1 expression is indicative of tumour ‘escape’ from the host immune response [98,99]. Bayraktar et al. illustrated that tumours with increased mutational burden are more likely to possess high levels of Ki-67 antigen expression [100]; these cancers are subsequently more likely to benefit from ICIs. Both Davey et al. and Ghebah et al. demonstrated a strong correlation between high PD-L1 expression with aggressive microscopic tumour features such as ER and PgR negativity, high grade, and increased Ki-67 expression within breast tumour cells [101,102,103,104]. Muenst et al. and Bae et al. reiterated these findings surrounding the correlation between increased Ki-67 expression and increased PD-L1 [105,106]. More recently, Asano et al. described increased PD-L1 expression to be associated with reduced Ki-67 [107], which is perhaps unsurprising as simple measures of PD-L1 expression does not capture differential enrichments across patients, tumour, and immune cell subtypes, as well as the spatial proximity of these cell types within tissues. Furthermore, these relational features may be critical for further evaluating the complex stimulatory and inhibitory processes that depend on the interplay between individual cells in the tumour microenvironment (TME). The bona fide validity of Ki-67 in predicting response to systemic and endocrine agents is evident in modern practice [85,108,109]; however, recent analyses suggest proliferative markers, including Ki-67, may be predictive of resistance to immune checkpoint inhibition in NSCLC [110]. On the contrary, a small pre-clinical trial of early-stage NSCLC patients described that Ki-67 expression correlates with increased immune checkpoint expression on both tumour and TILs within the TME [111], advocating that evaluating pre-treatment Ki-67 levels may present predictive value for those indicated to undergo combined cytotoxic chemotherapy, ICI, or novel combinations. Thus, it is imperative that the scientific community delve further into the relatability of Ki-67 expression in invasive breast cancers as a biomarker of responses to targeted therapies to inform therapeutic decision making in future practice.

### 2.3. Ki-67 and Multigene Panels

Contemporary oncology has advanced in concordance with our increased appreciation of genetic properties and application of genomics in cancer management [112]. Several genetic signatures have been developed to assist clinicians in personalising therapies specifically to each patient on the basis of the molecular properties of their disease [113]. In the management of breast cancer, genomic tools have been revolutionary in subtyping the molecular properties of breast cancer, guiding therapeutic decision making and predicting disease recurrence [114]. Multigene panels, such as the OncotypeDX^®^ 21-gene recurrence score (Genomic Health Inc., Redwood City, CA, USA) (RS), have been recognised by major oncology societies such as ASCO, National Comprehensive Cancer Network (NCCN),European Society of Medical Oncology (ESMO), National Institute for Health and Care Excellence (NICE) and St. Gallen Consensus panel, all of whom have incorporated the 21-gene assay into their guidelines [22,115,116,117,118], allowing RS to influence multidisciplinary decision making in well-resourced healthcare systems [119]. Within the multigene assay, a comparison between 16 cancer-related genes and 5 reference (or ‘house-keeping’) genes are made, generating a RS indicative of the likelihood of disease recurrence. Of the 16 cancer genes in the panel, five are directly related to proliferation, with one corresponding with Ki-67 antigen expression [120] (Figure 1). In recent times, there has been a critical vogue surrounding the degree of discordance between pathological parameters such as nuclear grade and Ki-67 indices alone/in isolation [121,122,123,124,125], rendering RS testing favourable in aiding prognosis, in spite of its limitations [126]. Therefore, it is somewhat unsurprising that the data from several studies highlight the correlative nature of RS and Ki-67 protein expression in Luminal breast cancer (*p* < 0.001) [114,127,128], particularly in cancers with high Ki-67 expression. In MammaPrint^®^ (Agendia, Amsterdam, The Netherlands), a 70-gene panel boasting comprehensive measurement of the six hallmarks of cancer-related molecular biology [129], their 70 genes were selected from genome-wide expression data using a data-driven approach in an unbiased fashion; there were no predefined assumptions as to whether certain genes were more likely to increase the risk of distant recurrence development in patients with early-stage breast cancer [130]. Despite Ki-67 being considered a practical biomarker of cancer proliferation, it was not included in the signature [131]; however, it has been proposed that Ki-67 may be a comparable biomarker to the 70-gene signature in guiding adjuvant therapeutic decision making (*p* < 0.001), which is unsurprising as increased Ki-67 is useful in predicting disease recurrence [70,132,133]. In recent times, long-term results of this prospective analysis involving almost 7000 patients diagnosed with node negative breast cancers or with 1–3 positive nodes suggests the 70-gene assay ‘de-escalates’ the requirement for adjuvant chemotherapy prescription in cases of low disease burden [134], while a prospective evaluation of RS testing in patients with 1–3 positive axillary nodes is underway in the treatment (Rx) for POsitive NoDe, Endocrine Responsive breast cancer (or RxPONDER, SWOG S1007) trial [135]. The correlation between the 21-gene and 70-gene signatures and Ki-67 indices remains explicit [70,127,132], with Pronzato et al. and Tian et al. presenting respective datasets of 305 and 1008 patients reinforcing such findings (*p* < 0.001). Therefore, the utility of Ki-67 in identifying groups of patients with ER+ disease is valid based on comparisons with the aforementioned “gold standard” multigene assays. Despite the reported limitations of Ki-67 as a consistent or independent marker to inform therapeutic decision making, its correlation with the RS and MammaPrint^®^ indicate its inescapable relevance in this space. Through the application of RS testing in the well-resourced healthcare economies of the western world, Ki-67 currently remains embedded into decision making in relation to cytotoxic chemotherapy. Moreover, the authors acknowledge that RS testing uses a polymerase chain reaction to evaluate Ki-67 expression; perhaps a routine assessment of the biomarker through these methods may improve standardisation and reproducibility of Ki-67 reporting in modern histopathological practice.

### 2.4. Improving Ki-67—Future Considerations

#### 2.4.1. Standardisation

As described in the current review, difficulty ascertaining a standard measure of Ki-67 across all breast carcinoma tissue has provided a challenge in histopathological evaluation. In their recent publication, Aung et al. present a novel methodology relating to the standardisation of an immunohistochemical cell line microarray (CMA) with TMA across 6 varying commercially available Ki-67 antibody clones [136]. Their results advocate TMA is capable of normalising the staining of these antibodies, with data validated across two Ki-67 expressing (Jurkat cells and Kaprass 299) and one Ki-67 negative (Sf9 (Spodoptera Frugiperda 9)) cell lines, as well as in a cohort of 109 TNBC patient samples. Briefly, using a bench-top protocol, the paraffin-embedded CMA and TMA slides were first de-paraffinised before the respective Ki-67 antigens were retrieved using citrate buffer (pH 6.0). Thereafter, each slide was incubated with the commercially available monoclonal Ki-67 antibodies at their recommended concentrations prior to incubating in hematoxylin and 3′-diaminobenzidine, with the aim to detect immunohistochemical reactivity. These results were standardised and validated in three independent laboratories at Yale University, and digital image analysis (DIA) was then performed to determine percentages of Ki-67 positive cells on stained slides; these means of standardisation between antibodies should prove promising in the quest to standardise Ki-67 staining for prospective histopathological tumour appraisal.

#### 2.4.2. Digital Image Analysis

The single most likely methodology to revolutionise current practice, eliminate the significant issue around heterogeneity, and produce clinically meaningful cut-offs is digital image analysis (DIA). In histopathology, DIA involves the processing of whole-slide digitalised images through microscopy and computer-based analyses to extract meaningful information that may inform histopathological reporting [137]. DIA has recently emerged as a reproducible and more accurate method of evaluating Ki-67 when compared to manual staining and scoring particularly over a large slide area [51,52,138]. While performing visual assessment (VA), 500–1000 cells must be included in order to obtain acceptable error rates and to correct for tumour heterogeneity (Figure 2) [52], with intra- and interobserver variability remaining a limitation. Using DIA methodology, such variability is less likely to impact the congruency of histopathological tumour appraisal for features such as Ki-67 expression due to the proposed algorithmic approach of DIA [128]. As previously outlined, the current clinical practice involves performing manual Ki-67 appraisal on whole tissue sections, as advocated by the International Ki-67 in Breast Cancer Working Group [52]. However, several studies in the literature have attempted to refine this robust methodology, and the data suggest considerable/robust concordance between manual scoring and DIA. Klauschen et al. describe promising results validating computer-assisted Ki-67 scoring in their analysis of over 1000 breast tumours [139], while Zhong et al. observed ‘almost perfect agreement between VA and DIA’ in high cases of increased Ki-67 expression, as well as a significant degree of homogeneity in staining among their 155 cases [140]. DIA based on virtual double staining (VDS) with fused parallel cytokeratin and Ki-67 (MIB1) has been described to be greater than 85% congruent with VA by Roge et al. in their evaluation of 140 core biopsies, further fuelling dispute as to the requirement for assessment of the whole tissue specimen [141]. In recent years, Stalhammar et al. have observed DIA outperforming VDS (with pancytokeratin CkMNF116 and Ki-67) in terms of sensitivity and specificity in differentiating Luminal A and B tumour molecular subtypes, as stratified by Prediction Analysis of Microarray 50 (or PAM50) [142]. Furthermore, VA and DIA matched one another in prognostication of HR for overall survival in tumours with high Ki-67 (defined as greater or equal to 20%) versus low Ki-67 expression. With promising results in support of DIA Ki-67 antigen evaluation, several other considerations must also be mooted. There can be an associated substantial investment to acquire digital pathology capacity as it is a disruptive technology for pathology laboratories. However, its incremental use as a means of enhancing precision medicine evaluation of biomarkers including Ki-67 as well as others such as PDL1 or Her2 would be less disruptive and likely to strengthen the use of Ki-67 as a clinically important biomarker. Thus, it is imperative that algorithmic techniques, such as those described by Karsnas et al., are integrated into proposed digital histopathology to ensure standardisation in reporting before the widespread adoption of this approach [143].

#### 2.4.3. Ki-67 and miRNA Analysis

Micro ribonucleic acids (miRNAs) are small, non-coding ribonucleic acids (RNAs) approximately 19–22 nucleotides in length and are known to regulate gene expression [144]. First described by Lee et al. in 1993 [145], miRNAs have a key role in cancer proliferation, with the clinical utility of prognostic, diagnostic and therapeutic avenues being explored through measuring miRNA expression profiles [146,147]. Increased Ki-67 correlates with aggressive, highly proliferative disease, and efforts have been made to augment Ki-67 indices through supplementation with miRNA expression data: Sakurai et al. performed hierarchical cluster analyses to elicit correlations between low, intermediate, and high levels of Ki-67 expression and miRNA expression [148]. Low Ki-67 expression was considered with scores of 0–14% (control group), and nine miRNAs were overexpressed in this group: miR-let-7a, miR-let-7b, miR-let-7e, miR-29a, miR-143, miR-181a, miR-214, miR-218, and small non-coding molecule, SNORD48. In the same analysis, almost 30 miRNAs were associated with high Ki-67 expression (scores of >25%), while two of the most significantly correlative of which were miR-191 (*p* = 0.080) and miR-7 (*p* = 0.051) (Table 2). Moreover, the expression of miR-let-7e is inversely correlated to Ki-67. This is unsurprising as the let-7 family is recognised as being involved in cancer differentiation [149]. Of note, Sakurai et al. illustrated miR-21, miR-96, and miR-125b to overlap into a group expressing increased HER2 positivity and Ki-67 [148]. On the contrary, miR-let-214 and miR-15a were expressed in low HER2-expressing cancers, as well as the low Ki-67 group, while miR-27a, miR-92a, miR-301a, miR-355a, and miR-16 were abundant within low HER2-expressing tumours, yet overexpressed in cancers with high Ki-67 expression. Amorim et al. evaluated the prognostic relevance of miRNA in patients diagnosed with Luminal breast cancers [150]. Following stratification for Ki-67 index, miR-30c-5p, miR-182-5p and miR-200-3p independently predicted endocrine resistance-free survival within this group, while miR-30c-5p (*p* = 0.005), miR-200b-3p (*p* = 0.003), and miR-182-5p (*p* = 0.001) were predictive of disease-free recurrence, once adjusted for Ki-67 status. These findings suggest that the application of these biomarkers combined in an array or independently with the Ki-67 index may be a clinically relevant approach to selecting patients at risk of endocrine resistance within Luminal disease. Finally, Liu et al. correlated miRNA with Ki-67 expression; the downregulation of miR-130b and miR-218, while the upregulation of miR-106b were all associated with Ki-67 expression [151]. Trang et al. describe the potential for the exploitation of Ki-67 as a miRNA target; mir-let-7 blockade suppressed Ki-67 levels in murine lung tumours; however, prognostication following such experimentation is limited given the paucity of subsequent data published [152]. At present, efforts to manipulate the relevant mRNAs involved in molecular pathways driving cancer proliferation have been limited, with a focus on Ki-67 and its associated miRNAs, which could be a potential avenue for future translational research.

#### 2.4.4. Ki-67 and Radiomic Analysis

Radiomics is an emerging translational field of research with the aim of extracting mineable high-dimensional data from clinical imaging, with the hope that these findings may aid diagnosis and assist in prognostication while guiding personalised therapeutic decision making [153]. Conventional cancer diagnosis and classification is based on histological evaluation of biopsied tissue; recent efforts have refocused diagnostics towards minimally invasive techniques, with radiomics emerging as a promising tool for precision medicine in cancer care [154]. While Ki-67 expression is measured from retrieved tumour tissue, the utility of radiogenomics in the identification of key tumour characteristics could facilitate the improvement of prognostication or prediction of therapeutic response, thereby informing therapeutic decision making in relation to neoadjuvant therapy. Juan et al. first described radiomic parameters (i.e., morphological tumour area, grey level skewness and kurtosis, grey level co-occurrence matrix contrast, correlation, homogeneity, inverse differential moment, etc., all *p* < 0.05) and their respective correlation with predicting Ki-67 indices in a series of 53 low Ki-67 (less than 14%), and 106 cases of high Ki-67 (greater to or equal to 14%) invasive breast cancers were evaluated using dynamic contrast enhanced magnetic resonance imaging (DCE-MRI) [155]. These findings imply preoperative tumour imaging may potentially allow for the prediction of the overall Ki-67 expression in a cancer, guiding respective neoadjuvant or adjuvant treatment decisions in a more cost and time efficient manner. Tagliafico et al. provided similar results using digital breast tomosynthesis imaging in their series of 70 women diagnosed with invasive breast carcinoma; tumour sphericity, autocorrelation (grey level co-occurrence matrix), interquartile range, robust mean absolute deviation, and short-run high grey-level emphasis all show an association with Ki-67 expression [156]. Ma et al. yielded similar results from their analysis using DCE-MRI, with previously described parameters, such as tumour area, skewness, kurtosis, and homogeneity, all correlating with Ki-67 indices [157], while Cui et al. have recently illustrated the clinical utility of ultrasound sonography in determining Ki-67 status [158]. These analyses highlight the opportunities presented through machine and deep learning radiomic techniques to further personalise medical treatment while promoting minimally invasive techniques where feasible. Moreover, the promising concept of radiogenomics (i.e., the clinical combination of radiologic phenotypes and molecular characteristics to aid cancer diagnostics and treatment) poses great potential in the augmentation of practical biomarkers, such as the Ki-67 proliferation indices [159,160]. Radiogenomics presents a novel opportunity to add further value to the clinical applicability of Ki-67, where a detailed appraisal of both radiomic and genomic data may aid the delineation of patients subgroups who may derive greater benefit from certain therapies, such as conventional chemotherapy prescribed in the neoadjuvant setting, as illustrated in Figure 3. As we enter the multiomic era, these encouraging advancements in the fields of genomic and radiomic medicine look certain to be at the forefront of future diagnostics, prognostication, as well as therapeutic decision making in breast cancer management, and provide the potential to enhance/augment the current value of Ki-67 proliferation indices in breast tumour histopathological and immunohistochemical appraisal.

## 3. Conclusions

Ki-67 proliferation indices provide precise measurement of the proliferative potential of breast cancer cells. Although widely utilised in histopathological evaluation, inconsistencies in the methodology of assessment, lack of gold standard guidelines, and varying uptake of multigene panels incorporating Ki-67 negatively impact the reliability and standardisation of this biomarker in clinical practice. Future practice may see digital image analysis, augmentation with microRNAs, or radiomic strategies attempt to enhance Ki-67 utilisation as a molecular biomarker within the breast cancer paradigm.

## Figures and Tables

**Figure 1 cancers-13-04455-f001:**
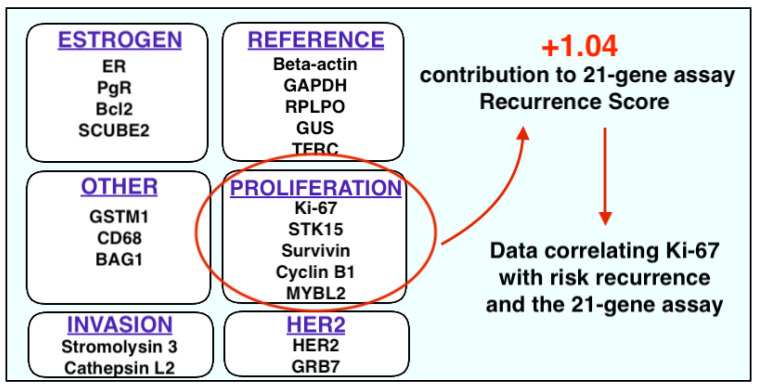
List of all genes assessed through the polymerase chain reaction in the OncotypeDX© 21-gene Recurrence Score signature (Genomic Health inc., Redwood City, CA, USA); proliferation contributes the largest proportion of included genes to the score with Ki-67 a key component of Ki-67 expression to the Recurrence Score through a number of techniques, including traditional immunohistochemistry [127] and novel machine learning techniques [114].

**Figure 2 cancers-13-04455-f002:**
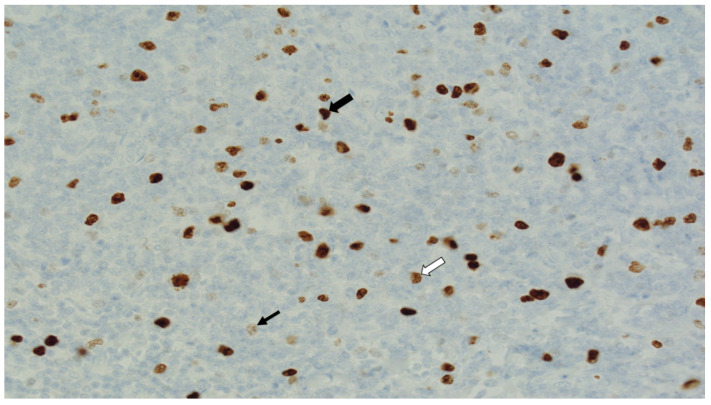
Figure demonstrating Ki-67 staining in lymphoid tissue illustrating the challenge required for digital image analysis in relation to assigning where the threshold for detection lies as evidenced by the differences in staining between cells from intense (black arrow) to intermediate (white arrow) to faint (thin arrow) (40× Magnification).

**Figure 3 cancers-13-04455-f003:**
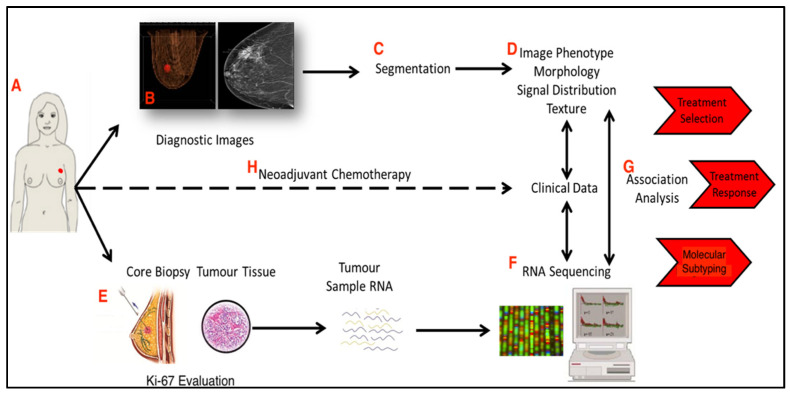
This figure depicts the systematic stages required to augment therapeutic decision making in relation to neoadjuvant chemotherapy in conjunction with Ki-67 evaluation using radiogenomic tumour appraisal. These stages begin at (**A**) initial presentation and are conducted through to combined analysis of clinicopathological, radiomic, and genomic data in order to personalise oncological care: (**B**) represents diagnostic preoperative imaging which is (**C**) segmented before quantitative data are retrieved from the acquired preoperative imaging (**D**). Histopathological data obtained from core tissue biopsy remain the ‘gold standard’ method of diagnosing malignancy (**E**); however, (**F**) molecular profiling of tumour tissue through RNA sequencing may be included in genomic data. Radiogenomics looks to collate clinical, radiomic, and genomic parameters through association analysis in order to better inform treatment selection, predict responses to therapies, and substratify disease subtypes and their associated prognoses (**G**). This schema illustrates the value of radiogenomics as an adjunct to enhance predicting the response to neoadjuvant chemotherapy (**H**) in the setting of breast carcinoma, with a focus upon utilising Ki-67 expression to aid this process.

**Table 1 cancers-13-04455-t001:** Studies assessing the validity of Ki-67 as a biomarker in invasive breast cancer.

Author	Year	N	Patients	Findings
Ellis [75]	2008	228	ER+ stage II/III	Per 2.7% increase in Ki-67 expression levels, there is an increased risk of RFS in patients treated with NET (HR: 1.3, 95% CI: 1.05–1.50)
Fasching [47]	2011	552	Early breast cancer	Using greater than 13% as a cut-off for Ki-67, Ki-67 predicted pCR ro NAC (OR: 3.5, 95% CI: 1.4–10.1) and OS (HR: 8.1, 95% CI: 3.3–20.4) and DDFS (HR: 3.2 95% CI: 1.8–5.9)
Brown [76]	2013	105	Received NAC	Ki-67 expression correlated directly to pCR
Niikura [77]	2014	971	ER+/HER2-	Patients with low Ki-67 expression indices had significantly better RFS and OS than those with intermediate- and high- Ki-67 expression (all *p* < 0.001)
Petrelli [69]	2015	64,196	All subtypes	In this meta-analysis, Ki-67 expression levels greater than or equal to 25% predicted OS in 64,196 breast cancer patients (HR: 2.05, 95% CI: 1.66–2.53)
Enrico [68]	2018	506	Stage I-III	Illustrated the 20% Ki-67 expression cut off as clinically relevant for recurrence and survival (HR: 7.14, 95% CI: 3.87–13.16)
Wu [74]	2019	7,716	Resected TNBC	In this meta-analysis, Ki-67 expression levels greater than 40% predicted DFS (HR: 2.30, 95% CI: 1.54–3.44) and OS (HR: 2.95, 95% CI: 1.67–5.19)
Zhu [27]	2020	1800	Early stage TNBC	Using a 30%, high Ki-67 indices independently predicted worse OS (HR: 1.947, 95% CI: 1.108–3.421)
Tian [70]	2020	1008	ER+/HER-	Ki-67 expression profiles correlated with the 70-gene assay; for patients with Ki-67 less than 15%, 81.4% were GLR

N; number, ER+; estrogen receptor positive, RFS; recurrence-free survival, NET; neoadjuvant endocrine agents, HR; hazards ratio, CI; confidence interval, pCR; pathological complete response, NAC: neoadjuvant chemotherapy, OS; overall survival, DDFS; distant-disease free survival, HER2-; human epidermal growth factor receptor-2 negative, RFS; recurrence-free survival, DFS; disease-free survival, TNBC; triple negative breast cancer, GLR; genetic low-risk following 70-gene signature stratification.

**Table 2 cancers-13-04455-t002:** Micro-RNA and their associations with Ki-67 proliferation index expression [144,146].

Author and Year	Country	Tissue	*N*	Technique	MicroRNA and Ki-67 Status
Sakurai 2018 [144]	Japan	Breast tumour	21	qRT-PCR	miR-let-7a, miR-let-7b, miR-let-7e, miR-29a, miR-143, miR-181a, miR-214, and miR-218 were all overexpressed in control breast cancer group, defined as possessing Ki-67 indices of 0–14% miR-7, miR-15b, miR-16, miR-18b, miR-20b, miR-21, miR-25, miR-27a, miR-27b, miR-34a, miR-92a, miR-96, miR-125a-5p, miR-125b, miR-132, miR-133b, miR-146a, miR-148b, miR-149, miR-150, miR-183, miR-184, miR-191, miR-199a-3p, miR-200c, miR-203, miR-301a, miR-355, and miR-363 were all upregulated in breast cancers with high Ki-67 expression (greater than 25%)
Amorim 2019 [146]	Portugal	Breast tumour	139	qRT-PCR	miR-30c-5p, miR-182-5p, and miR-200-3p expression profiles independently predict endocrine resistance-free survival once adjusted for Ki-67 status miR-30c-5p, miR-200b-3p, and miR-182-5p levels independently predict endocrine resistance-free survival once adjusted for Ki-67 status Predictive of disease-free recurrence, once adjusted for Ki-67 status

*N*; number, qRT-PCR: quantitative real-time polymerase chain reaction.

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
