# Peer review of "Ki-67 as a Prognostic Biomarker in Invasive Breast Cancer"

_cancers, 2021, doi:10.3390/cancers13174455_

Round 1

Reviewer 1 Report

Comments

  • The title is misleading, it indicates the review will discuss multiple biomarkers for invasive breast cancer, however only one biomarker is discussed, Ki-67
  • Page 2, line 62; “Furthermore, discovery and development of diagnostic, prognostic and therapeutic biomarkers have transformed the international perception such that at least four heterogeneous molecular subtypes are recognised in clinical practice (11).” the original St Gallen manuscript form Annals of Oncology 2011 (reference #18) would be a more appropriate reference for this sentence
  • Page 2, line 75; “Additionally, Ki-67 75 proliferation indices remain critical in the 2011 St. Gallen Consensus for differentiating Luminal molecular subtypes (24).” This sentence would be clearer if it was changed to “Additionally, Ki-67 75 proliferation indices remain critical in the 2011 St. Gallen Consensus for differentiating Luminal A and Luminal B molecular subtypes (24).”
  • Page 2, line 75; “Additionally, Ki-67 75 proliferation indices remain critical in the 2011 St. Gallen Consensus for differentiating Luminal molecular subtypes (24).” The incorrect reference has been used for this sentence it should be reference #18.
  • Page 6, line 283; “The bona fide validity of Ki-67 in predicting response to systemic and endocrine agents is evident in modern practice (86, 87, 89, 108, 109)” references 86, 89 appear to be used as an incorrect reference for this sentence as they do not mention Ki-67
  • The authors should include a table which compares the studies that assessed the validity of Ki-67 as a biomarker in invasive breast cancer
  • Table 1 is not clear and is difficult to read, it needs to be re-formatted
  • A systematic review and meta-analysis titled “Prognostic value of different cut-off levels of Ki-67 in breast cancer: a systematic review and meta-analysis of 64,196 patients” was published in 2015 in Breast Cancer Res Treat, this manuscript should be referenced
  • assessing prognostic value

Minor corrections

  • More subheadings would improve the readability
  • Page 3, 4 and 9; Both “Ki67” and “Ki-67” have been used, the naming should be consistent throughout the manuscript
  • Page 3, Line 148; KI-67 should be Ki-67
  • It would be good if an example of Ki-67 staining in a breast cancer tissue is shown, e.g., to compare low vs high Ki-67 expression

Author Response

REVIEWER 1 COMMENTS:

REVIEWER 1 COMMENT:

  • The title is misleading, it indicates the review will discuss multiple biomarkers for invasive breast cancer, however only one biomarker is discussed, Ki-67

AUTHORS RESPONSE:

Thank you for this suggestion – the authors have amended the title of this manuscript to ‘Ki-67 as a Prognostic Biomarker in Invasive Breast Cancer’. We hope this is satisfactory.

REVIEWER COMMENT:

  • Page 2, line 62; “Furthermore, discovery and development of diagnostic, prognostic and therapeutic biomarkers have transformed the international perception such that at least four heterogeneous molecular subtypes are recognised in clinical practice (11).” the original St Gallen manuscript form Annals of Oncology 2011 (reference #18) would be a more appropriate reference for this sentence

AUTHORS RESPONSE:

Thank you for this suggestion. We have amended the text such that the original St Gallen manuscript by Goldhirsch et al. 2011 is referenced as suggested.

REVIEWER COMMENT:

  • Page 2, line 75; “Additionally, Ki-67 proliferation indices remain critical in the 2011 St. Gallen Consensus for differentiating Luminal molecular subtypes (24).” This sentence would be clearer if it was changed to “Additionally, Ki-67 75 proliferation indices remain critical in the 2011 St. Gallen Consensus for differentiating Luminal A and Luminal B molecular subtypes (24).”

AUTHORS RESPONSE:

Thank you for this suggestion – we have amended the text accordingly such that it reads:

‘Additionally, Ki-67 75 proliferation indices remain critical in the 2011 St. Gallen Consensus for differentiating Luminal A and Luminal B molecular subtypes.’

REVIEWER COMMENT:

  • Page 2, line 75; “Additionally, Ki-67 proliferation indices remain critical in the 2011 St. Gallen Consensus for differentiating Luminal molecular subtypes (24).” The incorrect reference has been used for this sentence it should be reference #18.

AUTHORS RESPONSE:

Thank you for highlighting this error – the correct reference (Goldhirsch et al (2011) has been added to this sentence.

REVIEWER COMMENT:

  • Page 6, line 283; “The bona fide validity of Ki-67 in predicting response to systemic and endocrine agents is evident in modern practice (86, 87, 89, 108, 109)” references 86, 89 appear to be used as an incorrect reference for this sentence as they do not mention Ki-67

AUTHORS RESPONSE:

Thank you for highlighting this. We have amended our reference section accordingly such that only appropriate and relevant references are included.

REVIEWER COMMENT:

  • The authors should include a table which compares the studies that assessed the validity of Ki-67 as a biomarker in invasive breast cancer

AUTHORS RESPONSE:

Thank you for this suggestion – we have made a new table (now Table 1) which outlines the clinical validity of Ki-67 expression in breast cancer as recommended.

REVIEWER COMMENT:

  • Table 1 is not clear and is difficult to read, it needs to be re-formatted

AUTHORS RESPONSE:

Thank you for this suggestion. We have revised the format of this table (now Table 2) to improve readability and interpretation. Thank you.

REVIEWER COMMENT:

  • A systematic review and meta-analysis titled “Prognostic value of different cut-off levels of Ki-67 in breast cancer: a systematic review and meta-analysis of 64,196 patients” was published in 2015 in Breast Cancer Res Treat, this manuscript should be referenced

AUTHORS RESPONSE:

Thank you for bringing this study to the attention of the authors – we have revised the text to include the work of Petrelli et al. and highlight the prognostic significance attached to this study. Thank you.

Furthermore, Petrelli et al. outlined the prognostic significance of Ki-67 expression levels greater than or equal to 25% for predicting mortality in their review of over 64,000 breast cancer patients (HR: 2.05, 95% CI: 1.66 – 2.53).’

The work of Petrelli et al is now also outlined in Table 1.

REVIEWER COMMENT:

Minor corrections

  • More subheadings would improve the readability

AUTHORS RESPONSE:

Thank you for this suggestion – we have added relevant subheadings to the section 2.2 in order to improve the overall readability of the manuscript. Thank you

REVIEWER COMMENT:

  • Page 3, 4 and 9; Both “Ki67” and “Ki-67” have been used, the naming should be consistent throughout the manuscript
  • Page 3, Line 148; KI-67 should be Ki-67

AUTHORS RESPONSE:

Thank you for bringing these errors to the attention of the authors. All references to Ki-67 have been made uniform in this manner throughout the manuscript. Thank you.

REVIEWER COMMENT:

  • It would be good if an example of Ki-67 staining in a breast cancer tissue is shown, e.g., to compare low vs high Ki-67 expression

AUTHORS RESPONSE:

Thank you for this suggestion. We have included a figure outlining the challenge for digital image analysis when assigning the threshold for Ki-67  detection.

Thank you for your review.

Reviewer 2 Report

It is a rather well described review article about Ki-67 as prognostic and predictive biomarkers. The following comments are addressed.  Minor corrections appear necessary.

1. Page 5, line 2, is the reference 78 correct as the reference of the meta-analysis of 35 independent studies of almost 8000 patients with TNBC?

2. Page 8, line 367: Is the name of journal Nature correct?

3. Page 8, line 374: What does the abbreviation CMA mean?

4. In page 8, the paragraph of 2.4.2 Digital image analysis, the description of DIA itself does not appear sufficient.  The paragraph is lacking in description how to conquer the problem of intratumoral hererogeneity and the problem where to score in such uneven distribution of Ki-67 positive cancer cells.

5. In page 8, line 418 and 419, Ki67 should be Ki-67.

6. In page 10, line 460, the work of Trang et al is not cited as reference.

7. In Table 1, as the third article (ref. 150), the study by Liu et al in 2013 is introduced. However, this article is not for breast cancer but for malignant astrocytomas, and should be omitted.

8. Table 1 is not well organized and should be presented more clearly.  The correspondence between microRNAs and Ki-67 status in breast cancer is still unclear from the Table in the present form.

9. In the paragraph of 2.4.4. Ki-67 and Radiomic analysis, the situation of Ki-67 is unclear.  Thea authors argue that radiomics is minimal invasive techniques that substitute for invasive biopsied histopathological evaluation.  Because Ki-67 is measured from biopsied specimens, the discourse of the authors that radiomics incorporating measurement of Ki-67 lead to less invasive therapeutic decisions is not very understandable.

9. The manner of description of References is incomplete or inappropriate:  e.g., ref. #2, #48, #75, #79, #88, #94, #95, #101, #111, #114, #115, #118, #125, #132, #134-#136, #146, #149, #154, #156. In some of them, pages are incomplete or unnecessary hyphen is present.

10. Page 14: References 26 and 27 are duplicated.

Author Response

REVIEWER 2 COMMENTS

REVIEWER’S COMMENT:

It is a rather well described review article about Ki-67 as prognostic and predictive biomarkers. The following comments are addressed.  Minor corrections appear necessary.

AUTHOR’S RESPONSE:

Thank you for taking the time to review out manuscript. We believe the manuscript has benefitted greatly from the reviews of all three reviewers and are very grateful for your time and effort. Thank you.

REVIEWER’S COMMENT:

  1. Page 5, line 2, is the reference 78 correct as the reference of the meta-analysis of 35 independent studies of almost 8000 patients with TNBC?

AUTHOR’S RESPONSE:

Thank you for highlighting this error – the correct manuscript is that of Wu et al. which has now been referenced correctly at the end of this statement. In their meta-analysis of 35 studies, Wu et al. outline the prognostic significance of Ki-67 expression levels in 7,716 patients treated with curative intent for triple negative breast cancer:

Wu Q, Ma G, Deng Y, et al. Prognostic Value of Ki-67 in Patients With Resected Triple-Negative Breast Cancer: A Meta-Analysis. Front Oncol. 2019;9:1068. Published 2019 Oct 17. doi:10.3389/fonc.2019.01068

REVIEWER’S COMMENT:

  1. Page 8, line 367: Is the name of journal Nature correct?

AUTHOR’S RESPONSE:

Thank you for highlighting this – the work of Aung and colleagues was published in Modern Pathology which is a part of the Nature publishing group, and not the Nature journal itself. We have therefore amended this sentence to read:

‘In their recent publication, Aung et al. present novel methodology relating to the standardisation of an immunohistochemical cell line microarray (CLM) with TMA across 6 varying commercially available Ki-67 antibody clones.’

Aung TN, Acs B, Warrell J, Bai Y, Gaule P, Martinez-Morilla S, et al. A new tool for technical standardization of the Ki67 immunohistochemical assay. Mod Pathol. 2021.

REVIEWER’S COMMENT:

  1. Page 8, line 374: What does the abbreviation CMA mean?

AUTHOR’S RESPONSE:

Thank you for highlighting this error. In our review, we abbreviated cell line microarray to CLM. However, in the study by Aung et al., they use the abbreviation CMA to abbreviate cell line microarray. We have amended such that our abbreviation is congruent with the work of the authors of this study (reference below).

Aung TN, Acs B, Warrell J, Bai Y, Gaule P, Martinez-Morilla S, et al. A new tool for technical standardization of the Ki67 immunohistochemical assay. Mod Pathol. 2021.

REVIEWER’S COMMENT:

  1. In page 8, the paragraph of 2.4.2 Digital image analysis, the description of DIA itself does not appear sufficient.  The paragraph is lacking in description how to conquer the problem of intratumoral heterogeneity and the problem where to score in such uneven distribution of Ki-67 positive cancer cells.

AUTHOR’S RESPONSE:

Thank you for this suggestion. We have amended the section in relation to digital image analysis (DIA) in accordance to the suggestions of the reviewer in order to enhance the quality of the section. These include providing a description of the basic principles of DIA, as requested by the reviewer. We have then progressed to outline the functionality of DIA to reduce how problematic intra-tumour heterogeneity may be for reporting histopathologists through conventional manual analysis:

The single most likely methodology to revolutionise current practice, eliminate the significant issue around heterogeneity and produce clinically meaningful cut-offs is digital image analysis (DIA). In histopathology, DIA involves the processing of whole slide digitalised images through microscopy and computer-based analyses to extract meaningful information which may inform histopathological reporting. DIA has recently emerged as a reproducible, and more accurate method of evaluating Ki-67, when compared to manual staining and scoring particularly over a large slide area. While performing visual assessment (VA), 500-1000 cells must be included in order to obtain acceptable error rates and to correct for tumour heterogeneity, with intra- and interobserver variability remaining a limitation. Using DIA methodology, such variability is less likely to impact the congruency of histopathological tumour appraisal for features such as Ki-67 expression, due to the proposed algorithmic approach of DIA. As previously outlined, current clinical practice involves performing manual Ki-67 appraisal on whole tissue sections, as advocated by the International Ki-67 in Breast Cancer Working Group.’

We hope this section is now satisfactory.

REVIEWER’S COMMENT:

  1. In page 8, line 418 and 419, Ki67 should be Ki-67.

AUTHOR’S RESPONSE:

Thank you for bringing these errors to the attention of the authors. All references to Ki-67 have been made uniform in this manner throughout the manuscript.

REVIEWER’S COMMENT:

  1. In page 10, line 460, the work of Trang et al is not cited as reference.

AUTHOR’S RESPONSE:

Thank you for highlighting this – we have cited the work of Trang et al. as a reference to this review.

Trang, P., Medina, P., Wiggins, J. et al. Regression of murine lung tumors by the let-7 microRNA. Oncogene 29, 1580–1587 (2010).

REVIEWER’S COMMENT:

  1. In Table 1, as the third article (ref. 150), the study by Liu et al in 2013 is introduced. However, this article is not for breast cancer but for malignant astrocytomas, and should be omitted.

AUTHOR’S RESPONSE:

Thank you for this suggestion – we have removed the work of Liu et al from the table (now Table 2) as suggested by the reviewer. Thank you.

REVIEWER’S COMMENT:

  1. Table 1 is not well organized and should be presented more clearly.  The correspondence between microRNAs and Ki-67 status in breast cancer is still unclear from the Table in the present form.

AUTHOR’S RESPONSE:

Thank you for bringing this to the attention of the authors. We have revised this table (now Table 2) in order to allow the reader gain an understanding of the correlation between miRNA expression levels and Ki-67 status.

REVIEWER’S COMMENT:

  1. In the paragraph of 2.4.4. Ki-67 and Radiomic analysis, the situation of Ki-67 is unclear.  Thea authors argue that radiomics is minimal invasive techniques that substitute for invasive biopsied histopathological evaluation. Because Ki-67 is measured from biopsied specimens, the discourse of the authors that radiomics incorporating measurement of Ki-67 lead to less invasive therapeutic decisions is not very understandable.

AUTHOR’S RESPONSE:

Thank you for this comment. Although it is absolutely fair to say that current Ki-67 expression is measured from histopathological biopsies, the hypothesis surrounding radiogenomics suggests that minute data in preoperative imaging may provide similar information which may influence patient prognostication or guide therapeutic decision making. At present, the authors acknowledge that this platform is in its infancy, and is a long way from replacing conventional histopathological analyses performed on retrieved tissue, however the potential of radiomics in identifying data correlating to proliferation (as measured by markers such as Ki-67) would prove crucial in further personalising oncological care, while minimising invasive procedures where possible.

Section 2.4 of this review looks to outline the ‘future considerations’ and directions used to enhance or alter the current limitations or drawbacks of Ki-67 profiling in current practice; we imply that there may be merit in investigating radiological machine learning techniques (i.e.: deep learning, conventional neural networking, etc) and their potential utility in predicting proliferation as measured by markers such as Ki-67 expression Ultimately, once the threshold of Ki-67 expression is surpassed in clinical practice, patients are substratified into the ‘actionable’ group, who are indicated to receive cytotoxic chemotherapy. What the authors suggest in this review is that perhaps refinement of the current hypothesis that Ki-67 expression and other clinicopathological data may be predictable using radiomics may be useful for prognostication or the prediction of response to neoadjuvant therapy.

We have revised this section to reflect this:

While Ki-67 expression is measured from retrieved tumour tissue, the utility of radiogenomics in the identification of key tumour characteristics could facilitate the improvement of prognostication or prediction of therapeutic response, thereby informing therapeutic decision making relating to neoadjuvant therapy.

REVIEWER’S COMMENT:

  1. The manner of description of References is incomplete or inappropriate:  e.g., ref. #2, #48, #75, #79, #88, #94, #95, #101, #111, #114, #115, #118, #125, #132, #134-#136, #146, #149, #154, #156. In some of them, pages are incomplete or unnecessary hyphen is present.

AUTHOR’S RESPONSE:

Thank you for highlighting this issue with the references. We have amended the reference section such that these errors have been corrected. Thank you.

REVIEWER’S COMMENT:

  1. Page 14: References 26 and 27 are duplicated.

AUTHOR’S RESPONSE:

Thank you for highlighting this error in our citations. Reference 27 has been removed.

Thank you for your thorough review of our manuscript.

Reviewer 3 Report

This review focus on the clinical utility and future strategies to augment Ki-27 proliferation indices in breast oncology. It is a well written manuscript that summarises adequately the current bibliography. I propose to be accepted for publication after the following minor changes: In section 2.4.3. Ki-67 & MiRNA Analysis should be corrected to miRNA. Furthermore, Table 1 must be more comprehensive to the reader with well defined limits.

This review focus on the clinical utility and future strategies to augment Ki-27 proliferation indices in breast oncology. It is a well written manuscript that summarises adequately the current bibliography. Even though, there are several studies concerning the prognostic and predictive potential of Ki-67 in breast cancer, this manuscript is a contribution to this scientific area due to the fact that presents the clinical utility and future strategies on the use of Ki-67 as a promising biomarker in breast cancer oncology.

I propose to be accepted for publication after the following minor changes: In section 2.4.3. Ki-67 & MiRNA Analysis should be corrected to miRNA. Furthermore, Table 1 must be more comprehensive to the reader with well defined limits. This table must be re-formatted. In addition, the authors should include a table which compares the studies that assessed the validity of Ki-67 as a biomarker in invasive breast cancer.

Author Response

REVIEWER 3 COMMEMNTS

REVIEWER’S COMMENT:

This review focus on the clinical utility and future strategies to augment Ki-27 proliferation indices in breast oncology. It is a well written manuscript that summarises adequately the current bibliography. Even though, there are several studies concerning the prognostic and predictive potential of Ki-67 in breast cancer, this manuscript is a contribution to this scientific area due to the fact that presents the clinical utility and future strategies on the use of Ki-67 as a promising biomarker in breast cancer oncology.

I propose to be accepted for publication after the following minor changes:

AUTHOR’S RESPONSE:

Thank you for taking the time to review our manuscript please see below an outline of how the manuscript has been amended to address your comments.

REVIEWER’S COMMENT:

In section 2.4.3. Ki-67 & MiRNA Analysis should be corrected to miRNA. Furthermore, Table 1 must be more comprehensive to the reader with well-defined limits.

I propose to be accepted for publication after the following minor changes: In section 2.4.3. Ki-67 & MiRNA Analysis should be corrected to miRNA. Furthermore, Table 1 must be more comprehensive to the reader with well-defined limits. This table must be re-formatted. In addition, the authors should include a table which compares the studies that assessed the validity of Ki-67 as a biomarker in invasive breast cancer.

AUTHOR’S RESPONSE:

Thank you for these suggestions – we have amended the title to read ‘2.4.3. Ki-67 & miRNA Analysis’ and have edited the Table (Table 2) to make the table more comprehensive and increase its overall readability, as suggested by the reviewer.

We have added another table, (now Table 1), which compares the studies that assessed the validity of Ki-67 as a biomarker in invasive breast cancer. We hope you will find this to be satisfactory.

Thank you once again for taking the time to review our manuscript.

Round 2

Reviewer 1 Report

REVIEWER COMMENT:

Page 6, line 283; “The bona fide validity of Ki-67 in predicting response to systemic and endocrine agents is evident in modern practice (86, 87, 89, 108, 109)” references 86, 89 appear to be used as an incorrect reference for this sentence as they do not mention Ki-67

AUTHORS RESPONSE:

Thank you for highlighting this. We have amended our reference section accordingly such that only appropriate and relevant references are included.

REVIEWER COMMENT:

The authors modified sentence on Page 6, line 262: “The bona fide validity of Ki-67 in predicting response to systemic and endocrine agents is evident in modern practice [91, 113, 114]”

References 113 and 114 described studies that used chemotherapy and not endocrine therapy, these references are not correct for this sentence. If these two references are removed, it would not be acceptable to describe KI67 as a bona fide predictor with just single reference.

  1. Wang R-X, Chen S, Jin X, Shao Z-M. Value of Ki-67 expression in triple-negative breast cancer before and after neoadjuvant chemotherapy with weekly paclitaxel plus carboplatin. Scientific Reports. 2016;6(1):30091.
  2. Mukai H, Yamaguchi T, Takahashi M, Hozumi Y, Fujisawa T, Ohsumi S, et al. Ki-67 response-guided preoperative chemotherapy for HER2-positive breast cancer: results of a randomised Phase 2 study. British Journal of Cancer. 2020;122(12):1747-53.

REVIEWER COMMENT:

A new table 1 has been added: there appears to be a typo in the second reference in the table

“Using greater than 13% as a cut-off for Ki-67, Ki-67 predicted pCR ro NAC” should the “ro” be “to”

Author Response

REVIEWER 1

REVIEWER COMMENT: 

Page 6, line 283; “The bona fide validity of Ki-67 in predicting response to systemic and endocrine agents is evident in modern practice (86, 87, 89, 108, 109)” references 86, 89 appear to be used as an incorrect reference for this sentence as they do not mention Ki-67

AUTHORS RESPONSE: 

Thank you for highlighting this. We have amended our reference section accordingly such that only appropriate and relevant references are included.

REVIEWER COMMENT:

The authors modified sentence on Page 6, line 262: “The bona fide validity of Ki-67 in predicting response to systemic and endocrine agents is evident in modern practice [91, 113, 114]”

References 113 and 114 described studies that used chemotherapy and not endocrine therapy, these references are not correct for this sentence. If these two references are removed, it would not be acceptable to describe KI67 as a bona fide predictor with just single reference.

  1. Wang R-X, Chen S, Jin X, Shao Z-M. Value of Ki-67 expression in triple-negative breast cancer before and after neoadjuvant chemotherapy with weekly paclitaxel plus carboplatin. Scientific Reports. 2016;6(1):30091.
  2. Mukai H, Yamaguchi T, Takahashi M, Hozumi Y, Fujisawa T, Ohsumi S, et al. Ki-67 response-guided preoperative chemotherapy for HER2-positive breast cancer: results of a randomised Phase 2 study. British Journal of Cancer. 2020;122(12):1747-53.

AUTHORS RESPONSE: 

 Thank you for this comment and suggestion. The reference to systemic agents was to encompass systemic chemotherapies. We have amended this sentence such that it reads:

“The bona fide validity of Ki-67 in predicting response to systemic chemotherapeutic and endocrine agents is evident in modern practice [91, 113, 114]”

We have therefore not amended the reference list as both reference 113 and 114 refer to the utility of Ki-67 in guiding response to systemic chemotherapies. Thank you.

REVIEWER COMMENT:

A new table 1 has been added: there appears to be a typo in the second reference in the table

“Using greater than 13% as a cut-off for Ki-67, Ki-67 predicted pCR ro NAC” should the “ro” be “to”

AUTHORS RESPONSE: 

Thank you for highlighting this typo – we have amended the text.
